# Fertilization using manure minimizes the trade-offs between biodiversity and forage production in agri-environment scheme grasslands

Edith Villa-Galaviz[1¤]*, Simon M. Smart[2], Susan E. Ward[3], Mariecia D. Fraser[4], Jane Memmott[1]

1 School of Biological Sciences, University of Bristol, Life Sciences Building, Bristol, United Kingdom, 2 NERC Centre for Ecology & Hydrology, Library Avenue, Bailrigg, Lancaster, United Kingdom, 3 Lancaster Environment Centre, Lancaster University, Lancaster, United Kingdom, 4 Institute of Biological, Environmental and Rural Sciences, Aberystwyth University, Pwllpeiran, Cwmystwyth, Aberystwyth, United Kingdom

¤ Current address: Ecological Network lab, Technical University of Darmstadt, Darmstadt, Germany
* edith.villagalaviz@bristol.ac.uk

**Data Availability Statement:** The plant-insect data has been previously published and is in the public domain (https://datadryad.org/stash/dataset/

## Abstract

A common practice used to restore and maintain biodiversity in grasslands is to stop or decrease the use of fertilizers as they are a major cause of biodiversity loss. This practice is problematic for farmers who need fertilizers to increase forage and meet the nutritional needs of livestock. Evidence is needed that helps identify optimal fertilizer regimes that could benefit biodiversity and livestock production simultaneously over the long-term. Here, we evaluated the impact of different fertilizer regimes on indicators related to both biodiversity (plant, pollinator, leaf miners and parasitoid Shannon-Weiner diversity, bumblebee abundance, nectar productivity and forb species richness), and forage production (ash, crude protein, ruminant metabolizable energy and dry matter). To this end, we used data from a grassland restoration experiment managed under four nutrient inputs schemes for 27 years: farmyard manure (FYM; 72 kg N ha$^{-1}$ yr$^{-1}$), artificial nitrogen-phosphorus and potassium (NPK; 25 kg N ha$^{-1}$ yr$^{-1}$), FYM + NPK (97 kg N ha$^{-1}$ yr$^{-1}$) and no-fertilizer. Results showed strong trade-offs between biodiversity and forage production under all treatments even in applications lower than the critical load in the EU. Overall, farmyard manure was the fertilizer that optimized production and biodiversity while 97 kg N ha$^{-1}$ yr$^{-1}$ of fertilizer addition (FYM+NPK) had the most negative impact on biodiversity. Finally, forage from places where no fertilizer has been added for 27 years did not meet the nutritional requirements of cattle, but it did for sheep. Rethinking typical approaches of nutrient addition could lead to land management solutions suitable for biological conservation and agriculture.

doi:10.5061/dryad.vmcvdncr3). All relevant data are within the manuscript and its Supporting Information files.

**Funding:** "Project was partly funded by Bristol Centre for Agricultural Innovation (BCAI, project no.42). https://www.bristol.ac.uk/biology/bcai/ EVG was funded by a grant from CONACYT, Mexico for graduate studies (381429). https://conacyt.mx SMS was supported in part by the UK-SCaPE program delivering National Capability (NE/R016429/1) funded by the Natural Environment Research Council. https://www.ukri.org/councils/nerc/" The funders had no role in study design, data collection and analysis, decision to publish, or preparation of the manuscript."

**Competing interests:** The authors have declared that no competing interests exist.

## Introduction

Intensive agriculture is a significant cause of the decline in biodiversity in general, and a known driver of insect declines [1]. This decline in insect populations threatens ecosystem services like pollination and pest control, which are essential for both biodiversity and agriculture [2–4]. Although, agricultural practices such as organic farming and agri-environmental schemes have been proposed and implemented, the uptake and efficacy of these options have not been sufficient to stop biodiversity loss [5]. This problem is partly due to gaps and biases in the evidence needed to achieve effective management and inadequate consideration of farmer's need to maintain a viable business [see 6–8].

Many grassland ecosystems have their origins in an ancient agriculture created by humans and their livestock and these habitats have become highly diverse under centuries of human management [9]. In the last century, traditional practices like addition of manures and low grazing levels have been replaced by modern agriculture that uses high levels of mineral fertilizers and reseeding with productive forage grasses to allow higher grazing rates [10]. This change has contributed significantly to widespread habitat and biodiversity loss, with the loss of up to 97% of the original coverage of seminatural grasslands in Europe [11]. This high loss in cover has led to a considerable interest in preserving the remaining areas and restoring areas under intensive agricultural practices. [12,13]. Most proposed restoration actions are wildlife-friendly practices such as: a) replacing application of mineral fertilizers with organic manures, b) stopping all fertilization, c) reducing grazing intensity and d) using livestock with lower nutritional needs [14–16]. Although, there is consensus that reducing grazing intensity is necessary [15,17,18] reducing nutrient inputs is more complicated, since fertilizers are not only applied to increase biomass but also to meet livestock nutritional needs [see 19,20]. Moreover, despite evidence that using hardy or native types of livestock (breeds) at low intensity can be an effective farming approach ([e.g. 21], it is far from being a widespread approach.

There is good evidence that increasing plant diversity affects livestock production positively; for example, it can increase yield [22] and forage from species-rich grasslands can be of greater nutrient value than cereals and conventional forage [23]. Likewise, trade-offs of managing land for conservation or production have been widely discussed under the idea of multi-functionality of ecosystems e.g., [24,25] and within the land sharing vs land sparing debate [26–28]. Nevertheless, there is little empirical evidence on how restoring and preserving grasslands using different fertilization regimes, including that of no-fertilizer, could benefit biodiversity and livestock production simultaneously.

Recent evidence shows that, at the landscape level, the productivity of grasslands and conservation of arthropods can potentially be improved without reducing production or jeopardizing conservation objectives [29]. Recent work also shows that the optimum for production and conservation can be potentially achieved by a combination of land sharing and land sparing strategies across the landscape [27,28]. Although, these studies offer important insights for maintaining biodiversity in agricultural systems, most of them measured productivity using yield without considering the fact that productivity also depends on the nutritional quality of forage e.g., [6].

Low levels of mineral and manures are allowed under agri-environmental schemes. Under such schemes, even low fertilizer loads can inhibit the establishment of target plant species, thus hindering the restoration of grasslands without a significant improvement in production [12]. Indeed the critical load for nitrogen in European low and medium altitude meadows was recently confirmed at 20–30 kg N ha$^{-1}$ yr$^{-1}$ a range above which increase in grass and loss of diversity is expected [30]. Nevertheless, subsides provided to address the loss of yield consistent with reduced inputs are sometimes insufficient to compensate for income forgone [26].

Here, we evaluate the impact of different nutrient input practices (no fertilizer addition, farm yard manure, mineral fertilizer (N:P:K) and farm yard manure plus mineral fertilizer) on indicators supporting both biodiversity and livestock production. We used information on four components of biodiversity: plants, pollinators, herbivores and parasitoids, and forage nutritional content collected in a restored grassland, managed under each of the four nutrient input regimes for 27 years. Our research objectives are as follows: (a) compare the performance of the four different fertilizer regimes on indicators relating to plant community, insect community, forage yield and forage nutrient content, (b) evaluate the trade-offs of each fertilizer practice on biodiversity and forage production, (c) investigate which fertilizer regime optimizes the indicators collectively, noting that this is not the same as achieving maximum values of each indicator, and (d) use our results to estimate the potential livestock production under each treatment.

## Materials and methods

### Field site and experimental design

The study was carried out at the Colt Park Hay Meadow Field Trial, a long-term nutrient manipulation experiment established in 1991, based in Ingleborough National Nature Reserve (North Yorkshire, England, grid reference SD775782; 54˚12'N, 2˚21'W). At the start of the experiment the grassland was dominated by the perennial grass species *Lolium perenne* and *Cynosurus cristatus*. The topsoil was a clayey brown earth over limestone bedrock (pH ~5.8; 8.9 C%; 0.92 N% [31]). The aim of the experiment was to test different management strategies for improving the plant species diversity of grasslands within farms. Strategies included seed addition during the first 4 years (1990–1994) and different nutrient inputs, including no input [12] for 27 years.

The experiment is managed as part of a working farming system typical in the region. Management involves sheep grazing from March to mid-May, application of the fertilizer treatments in late May, hay cutting after 21 July, and sheep grazing for a further two weeks after the hay-cut, followed by cattle grazing during early winter. Fertilization application is done following the maximum loads allowed by agri-environmental schemes. It consists of the addition, once per year, of the following four treatment of nutrient inputs: (1) farmyard manure (12 t ha$^{-1}$ yr$^{-1}$, with an average content of 25 kg of dry matter (300 kg ha$^{-1}$ yr$^{-1}$) and nutrient composition of 6 kg of nitrogen (72 kg ha$^{-1}$ yr$^{-1}$), 3.5 kg of phosphate (42 kg ha$^{-1}$ yr$^{-1}$) and 8 kg of potash (96 kg ha$^{-1}$ yr$^{-1}$) per tonne of which only 20% are available for plants following application [32]), hereafter FYM; (2) low levels of mineral fertilizer (in the ratio 20 N:10 P:10 K, 25 kg ha$^{-1}$ yr$^{-1}$ N plus 12.5 kg ha$^{-1}$ yr$^{-1}$ of $P_2O_2$ and $K_2O$) hereafter referred to as NPK; (3) both fertilizers together (FYM+NPK equivalent to 97 kg ha$^{-1}$ yr$^{-1}$ N, 54.5 kg ha$^{-1}$ yr$^{-1}$ of $P_2O_2$ and 108 kg ha$^{-1}$ yr$^{-1}$ $K_2O$); and (4) a no-fertilizer control, hereafter no-fertilizer. There were six replicates of each treatment, each 2.5 m x 6 m (15 m$^2$) in size, arranged in three blocks. In each block, the six plots were randomly assigned to each fertilizer treatment and the same fertilizer treatment was applied once per year for 27 years.

### Data collection

We collated data on the effect of the four treatments from the 27-year field experiment on 12 indicators covering plant communities, insect communities and forage in terms of yield and nutrient content (Table 1). These indicators proved to be independent as no correlations equal or higher to 90% were seen among variables (Table 1 in S1 File). We set a threshold of 90% due to the fact that correlations between variables are expected as they respond jointly to fertilization [33]. Setting such a threshold allow us to detect variables that are not independent. For

**Table 1. Individual indicators describing the plant community, insect community and forage.**

| Indicator | Description |
|---|---|
| **Plant community** | |
| Diversity of plants | Shannon-Weiner index. |
| Forb species richness | Number of forb plant species |
| Nectar productivity | Estimated µl of nectar produced by the total number of forbs present in each plot. |
| **Insect community** | |
| Abundance of *Bombus* spp. | Number of observations of bumblebees feeding on the plots. |
| Diversity of herbivores | Shannon-Weiner index |
| Diversity of parasitoids | Shannon-Weiner index |
| Diversity of pollinators | Shannon-Weiner index |
| Percent parasitism | Proportion of emergences parasitoids from the total leaf miner emergences. |
| **Forage** | |
| Ash | Organic matter content. |
| Crude protein | Measurement of nitrogen supply from the forage. |
| Ruminant metabolizable energy | Energy available for supporting metabolic processes and for growth or reproduction after accounting for losses in digestion, gases and urine. |
| Mean dry matter (2011–2014) | Yield of biomass per m$^2$ following drying to constant weight. |

plant communities, we compared plant Shannon-Weiner diversity index, forb species richness and nectar productivity. For insect communities we compared the abundance of bumblebees, the Shannon-Weiner diversity index of pollinators, leaf miners and parasitoids, and the percentage of herbivores attacked by parasitoids. For forage we compared the mean dry weight of forage in the years 2011–2014 as a measure of yield; and three measures of nutrient content: ash, crude protein and ruminant metabolizable energy. These indicators were based on information from the following surveys: a vegetation survey from 2014 [34] and forage productivity data for the period 2011–2014 [35]; insect and flower surveys from 2016 [33]; and field sampling of forage collected for this study in 2017. While these datasets were gathered in different years, they were all collected towards the end of a 27-year experiment known to have exerted strong, detectable cumulative impacts on the vegetation [33], and so differences across treatments were expected to be larger than year-to year variations. Data collection and indicator descriptions are as follows:

## Plant community

For plant diversity and forb species richness: we used data from an existing vegetation survey, that recorded plant species and percent cover within 2 x 2 m quadrats [34]. Information on the number of species and on percent-cover were then used to estimate the Shannon-Weiner diversity index.

For nectar productivity, we combined data on flower abundance from 2016 [33] and nectar quantity per species from literature [36]. Flower abundance was collected by counting all flowers within three transects of 30 cm x 5 m lengthwise. Sampling took place in 2016 during late May to early June and again 3 weeks later for a total of two rounds. To estimate nectar productivity, we multiplied the amount of nectar produced by one flower of each of the species by the total number of flowers of the species found in the plot. We then calculated the total amount of nectar for each plot by summing the amount of nectar produced by all the species recorded in each plot.

## Insect community

Insect surveys were performed at the same time as the flower abundance survey. Leaf miners were collected from the same transects used for the flower abundance. Leaf miners were reared individually until an adult leaf miner, or a parasitoid emerged. Pollinators were collected using a hand net for 8 min in each plot between 09:00 and 17:00 hrs and sampling was repeated three weeks later for a total of three rounds (for a total of c.576 min). The level of sampling effort was deemed adequate since the species counts recorded 74% of the pollinator species, 97% of the herbivores and 85% of the parasitoid species estimated to occur in the Colt Park experiment (see 33). Here, we used information for the 24 plots or 33% of plots where no seed were added (see S1 File section 1). The dataset includes information records for 62 species of pollinators (n = 352) including 4 species of bumblebees (n = 15), 25 species of herbivores (n = 1565) and 22 species of parasitoids (n = 515). Using this information we calculated the Shannon-Weiner diversity index for each of the three insect groups. Finally, we calculated percent parasitism of the herbivores.

## Forage

Mean forage productivity was measured from hay cut at peak biomass over a four year period (2011–2014). The samples were collected from the center of the plot (1 m from the plot edge), dried in an oven at 60˚C for 48 hours and weighed.

For the nutrient content of forage, we collected samples in 2017. We used three transects of 5m separated by 50 cm in each plot, and a sample of c. 20 g of plant material per meter, was clipped to ground level along each transect. The samples were combined to provide a total of 300g of hay per plot. Each sample was then oven dried for 48 hrs. at 65˚C and analyzed for ash, crude protein, neutral cellullase, gammanase digestibility and ruminant metabolizable energy by Sciantec Analytical services (http://www.sciantec.uk.com/services.php?service=forage).

## Statistical methods

**Comparing the performance of four different fertilizer regimes with respect to indicators relating to plant community, insect community, forage production and forage nutrient content.** To compare the performance of each fertilizer treatments for each indicator and group of indicators (group average), we compared normalized mean values of the indicators from each group of indicators (plants, insects and forage). To this end, we first we standardized all indicators by subtracting the mean of each variable and dividing it by its standard deviation, this providing values in the same scale for all indicators while retaining the distributional features of each indicator measured. Finally, we estimated the performance of each fertilizer across all indicators by estimating the weighted mean of the 12 indicators (i.e. a group mean). We calculated a weighted mean to avoid any bias due to unequal number of indicators representing forage and biodiversity. The mean values for each indicator were weighted considering that all indicators related to forage should contribute to 50% of the performance of each fertilizer while indicators related to biodiversity should contribute to the other 50% (with indicators relating to plant communities contributing 25% and indicators of insect communities contributing the other 25%). Finally, to compare performance for each group of indicators individually, we also calculated the arithmetic mean for each group of indicators (forage, plant community and insect community).

**The effect of fertilization regimes on the trade-offs between the plant community, insect community and livestock forage.** Because of the relatively low number of replicates (n = 6 per treatment) we used randomization tests for differences among treatments for each of the 12 indicators. We also used the same approach to test for differences among the groups

of indicators (i.e. plant community, insect community and forage). Randomization tests compare observations to a probability distribution based on randomizations of the observed values under the single assumption that all randomizations have the same probability to be selected [37]. These tests do not rely on parametric assumptions regarding the distribution of the data, and so are especially useful for complex designs and low number of samples that reduce statistical power [38]. Hypothesis testing is similar to other statistical tests with the only difference that the probability value is estimated using the distribution created with the randomized observations. This type of analysis is widely used in ecology including estimation of species importance and ecosystem function [39,40].

Randomization was done as follows; we randomly shuffled the values of each indicator within blocks, but between treatments to retain the spatial correlation structure of the data. We then randomly selected one value for a pair of treatments and calculated the difference in mean between the two treatments carrying out a total of 54 comparisons considering the possible treatments combinations and the three blocks design. We then repeated the process 10,000 times to build a reference distribution. Finally, the observed mean difference between the same pair of fertilizer regimes was compared to the reference distribution. The P value represents the proportion of times the value of each observed test statistic was greater or equal to each randomized test statistic. Thus significant P values fall within the 97.5 or the 2.5% tails of the reference distribution, leading us to reject the null hypothesis of no difference between the observed and randomized test statistics for the pair of treatments. In practice, we use the absolute difference between the observed and randomized statistic so that P values are deemed statistically significant if $< = 0.05$. This process was repeated for all the fertilizer treatment combinations, so all fertilizers were compared between each other. The randomization testing scheme is illustrated in Fig 1 in S1 File for one of the three experimental blocks and for one pair of treatments only. Analyses used the vegan [41], dtplyr [42], plyr [43] ggplot2 [44] and Hmisc [45] packages embedded in a workflow written in R version 4.1.3 [46]. The R code is provided as supplementary information (S2 File)

**Identification of the optimum fertilizer regime supporting forage, plant communities and insect communities.** Fertilization drives trade-offs in grasslands. This is because increasing productivity drives changes in the plant community that ultimately affect positively or negatively insect communities [33]. Therefore, we defined the optimum fertilizer regime not only as the one with the highest mean value across all the indicators from all group of indicators but also the one with the lowest variation across all indicators. In this sense, the optimum regime is the one that minimizes the trade-off between group averages and maximizes their average values. As a result, the range of the indicators associated with the optimum treatment may well be greater than the observed maximum or minimum of any one indicator observed across the experiment (Fig 2 in S1 File).

We again used randomization tests to assess for differences between treatments. However, instead of comparing mean differences for each indicator individually. We built distributions and compared differences between pair of treatments for the weighted mean and weighted standard deviation calculated across all indicators (group average); that is instead of the mean being derived from one indicator we calculate a weighted mean across all 12 indicators. In both cases, statistical weighting was established as explained in the first section. We repeated the process for each pair of fertilizers and estimated P values as explained in the previous section. However, for weighted standard deviation a significantly lower observed indicator is interpreted as better performance because it indicates that the mean is derived from indicators that vary less across the optimum treatment.

### Estimation of the number of livestock that can be supporting by the different fertiliser treatments

We estimated the number of livestock that could be fed with forage produced under each type of fertilizer application regime. We consider both the amount of forage produced (yield) and the nutritional content of the forage. Because plots in the experiment are too small to support livestock, we used information from the literature to make estimations. We did this for Limousin suckler cows, assuming an average weight of 600kg per cow [47]; these cattle are mainly bred for meat production and breeding stock in our experiment as well as in upland grassland systems in England more generally [48]. Given recommendation for farmers on hay nutrient content and food requirement of livestock are reported per hectare in in the literature [49,50], we extrapolated the yield for each of our plots, to that from one hectare [see 12,35,51]. We, then, used the software FarmIQ (2016) to calculate the forage required for feeding cows over winter ahead of spring calving [49], while adjusting the equation parameters to reflect the measured metabolizable energy (ME) content of the forage. Based on the calculated allowance of forage, a spring-calving suckler cow with a body weight of 600 kg would require an allowance of 9.2 kg DM/d (dry matter per day); see S1 File Section 2 for the details of this calculation.

We also calculated the nutritional value of the forage available (i.e. metabolizable energy content) to sheep, another livestock animal commonly kept in the uplands [48]. We followed the same method used for the estimation of cattle production but considering that the equivalent forage allocation for mixed aged ewes with an average scanning percentage of 150% and due to lamb in mid-April is 1.4 kg DM/d of forage. Scanning percentage is a performance measure used to tailor management (i.e., nutritional needs) and production (lambing), e.g., [52] and in this case assumes half the ewes will be carrying single lambs, and half twins [47].

## Results

### Comparing the performance of the four different fertilizer regimes with respect to indicators relating to plant community, insect community, forage production and forage nutrient content

When comparing different indicators we observed that there was no single fertilizer treatment that had a constant performance across indicators, with treatments showing high and low values for indicators within and between forage production, plant communities and insect communities. Moreover, all treatments had a weighted mean of approximately 0.5, showing strong trade-offs between biodiversity and forage production under all treatments (Fig 1). Nevertheless, FYM showed the highest weighted mean (0.52) followed by no-fertilizer (0.47) and NPK (0.46) and NPK + FYM (0.46). The values of individual indicators are reported in Table 2 in S1 File.

### The effect of fertilization regimes on the trade-offs between the plant community, insect community and livestock forage

For forage related indicators, we observed lower crude protein in the no-fertilizer control compared to FYM, NPK and NPK +FYM treatments, and lower values in NPK-only compared to NPK+FYM. Similarly, we observed lower ruminant metabolizable energy in no-fertilizer than in FYM and NPK treatments compared to FYM. Finally, mean dry content of hay 2011–2014 was lower in the no-fertilizer control than in FYM, NPK and NPK+FYM. Although, not significant at the 5% level, the probability that FYM had a higher crude protein content in forage than NPK was 93% while the probability that FYM has a higher mean dry content was of 93%. We observed no significant differences in ash content among fertilizers. When comparing the

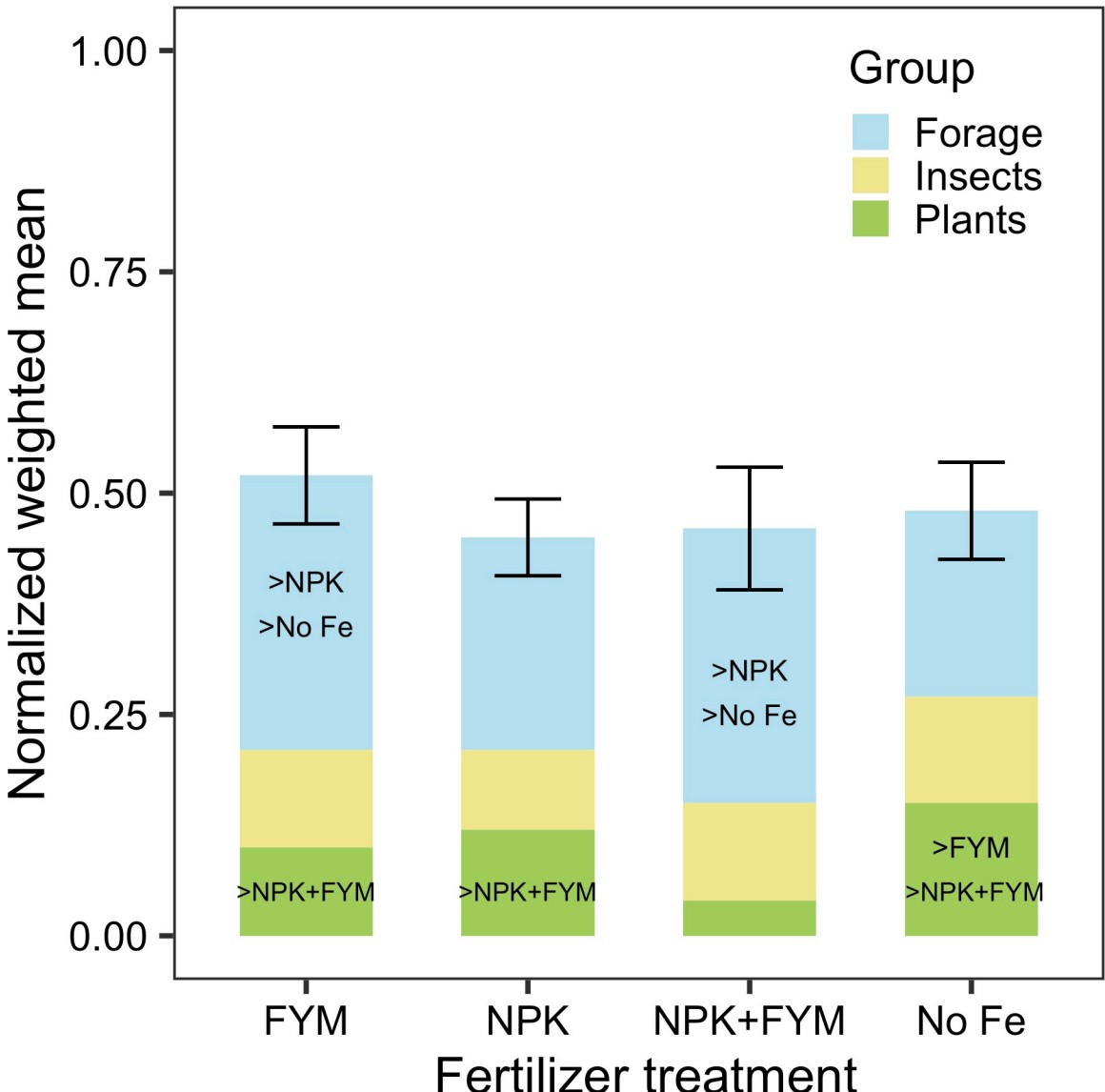

**Fig 1. The weighted mean of the 12 indicators for each fertilizer treatment.** Values are standardized and correspond to values between 0–1. Values were weighted considering that forage indicators contribute 50% of the mean value, insect community indicators (25%) and plant community indicators (25%). Colours correspond to the proportion that each group of indicators contribute to the weighted mean value. *No Fe*: no-fertilizer. NPK: mineral fertilizer (20:10:10), FYM: farm yard manure, NPK+FYM: mineral fertilizer and farm yard manure. Bars are mean ± standard error. Significant higher group average of a treatment vs other treatment, are indicated inside the bar of the treatment which has a higher mean.

group average, FYM and NPK+FYM showed a higher mean performance than NPK (P < 0.05) and no-fertilizer (P < 0.05).

For indicators related to the insect community, we observed a significantly higher abundance of bumblebees in the no-fertilizer control than in NPK and NPK+FYM; and in NPK compared to NPK-FYM. In the case of percent parasitism, FYM showed higher values than the no-fertilizer control and NPK. We observed no significant differences for diversity of herbivores, diversity of pollinators and diversity of herbivores among the different fertilizer treatments and there were no significant differences among treatments when comparing the group average.

**Table 2. Pairwise comparisons of the weighted mean between fertilizer treatments for each of the 12 indicators.**

| | Probability of weighted mean > weighted mean of no-fertilizer | | | | | | | | | | | |
|---|---|---|---|---|---|---|---|---|---|---|---|---|
| | *Forage* | | | | *Insect community* | | | | | *Plant community* | | |
| | Ash | Cp | Mw | Re | Ba | Dh | Pp | Dpa | Dpo | Dpl | Fs | Np |
| FYM | 0.31 | 0.99* | 0.99* | 0.99* | 0.12 | 0.55 | 0.99* | 0.34 | 0.2 | 0.14 | 0.07 | 0.13 |
| NPK | 0.38 | 0.98* | 0.99* | 0.17 | 0.03* | 0.12 | 0.2 | 0.46 | 0.47 | 0.37 | 0.22 | 0.16 |
| NPK+FYM | 0.46 | 0.99* | 0.97* | 0.74 | 0* | 0.62 | 0.81 | 0.82 | 0.13 | 0.11 | 0* | 0* |
| | Probability of weighted mean > weighted mean of farmyard manure | | | | | | | | | | | |
| No Fe | 0.69 | 0.004* | 0.01* | 0.01* | 0.87 | 0.45 | 0.02* | 0.65 | 0.79 | 0.86 | 0.93 | 0.86 |
| NPK | 0.53 | 0.07* | 0.07 | 0.01* | 0.5 | 0.17 | 0.03* | 0.61 | 0.78 | 0.73 | 0.83 | 0.6 |
| NPK+FYM | 0.59 | 0.53 | 0.63 | 0.22 | 0.25 | 0.57 | 0.27 | 0.87 | 0.4 | 0.32 | 0.001* | 0.05* |
| | Probability of weighted mean > weighted mean of NPK (20:10:10) | | | | | | | | | | | |
| No Fe | 0.62 | 0.02* | 0.001* | 0.83 | 0.97* | 0.88 | 0.8 | 0.54 | 0.53 | 0.63 | 0.78 | 0.85 |
| FYM | 0.46 | 0.93* | 0.93 | 0.99* | 0.5 | 0.83 | 0.97* | 0.39 | 0.2 | 0.26 | 0.16 | 0.4 |
| NPK+FYM | 0.56 | 0.96* | 0.87 | 0.91 | 0* | 0.85 | 0.92 | 0.81 | 0.18 | 0.16 | 0* | 0.01* |
| | Probability of weighted mean > weighted mean of NPK+FYM | | | | | | | | | | | |
| No Fe | 0.54 | 0.004* | 0.02* | 0.26 | 1* | 0.37 | 0.19 | 0.18 | 0.86 | 0.88 | 1* | 1* |
| FYM | 0.41 | 0.47 | 0.37 | 0.78 | 0.57 | 0.43 | 0.73 | 0.13 | 0.62 | 0.67 | 0.99* | 0.95* |
| NPK | 0.44 | 0.04* | 0.12 | 0.08 | 1* | 0.15 | 0.08 | 0.19 | 0.81 | 0.84 | 1* | 0.99* |

The probabilities correspond to the number of times out of the 10,000 randomizations, that the mean of the first treatment was higher than the mean of the second treatment. We set the two-tailed probability of significant difference as 0.1 (P < 0.05 or >0.95). * indicates significant differences. Forage: Re: Ruminant metabolizable energy. Mw: Mean dry weight 2011–2014. Cp: Crude protein. As: Ash. Insect community: Pp: Percent of parasitism. Dpo: Shannon-Weiner diversity of pollinators. Dpa: Shannon-Weiner diversity of parasitoids. Dh: Shannon-Weiner diversity of herbivores. Ba Abundance of bumblebees. Plant community: Np: nectar productivity. Fs: Forb species richness. Dpl: Diversity of plants. No Fe: No-fertilizer. NPK: mineral fertilizer (20:10:10), FYM: farm yard manure, NPK+FYM: mineral fertilizer and farm yard manure. N = 24.

Finally, for indicators related to the plant community, NPK+FYM had lowest forb species richness among all treatments. Similarly, NPK+FYM had the lowest nectar productivity among treatments. While not significant at the 5% level, the probability of FYM having a lower forb species richness than the no-fertilizer control was 93%. We did not observe significant differences in diversity of all plants among treatments. When comparing the group average, FYM and NPK+FYM showed a lower mean performance than no-fertilizer (P < 0.05). Meanwhile NPK+FYM had a significantly lower group mean for plant community indicators than FYM and NPK (P < 0.05). All comparisons and P-values for individual indicators are available in Table 2 and for groups average in Table 3 in S1 File. Standardized range of values of each indicator are shown in Fig 2.

## Identification of the optimum fertilizer regime supporting forage, plant communities and insect communities

When comparing the weighted mean between fertilizers, FYM was the only treatment that showed significant differences with some other treatments (Fig 3, Table 4 in S1 File). Thus, FYM had a significantly higher weighted mean than the control (P = 0.03) and NPK (P = 0.03), although there was no significant difference to NPK+FYM (P = 0.1). However, when comparing the weighted standard deviation among treatments, FYM has significant lower weighted standard deviation than NPK +FYM (P = 0.008). In fact, NPK+FYM showed a significantly higher weighted standard deviation than the rest of the treatments (P < 0.05). No significant differences in weighted standard deviation were observed among FYM, NPK and

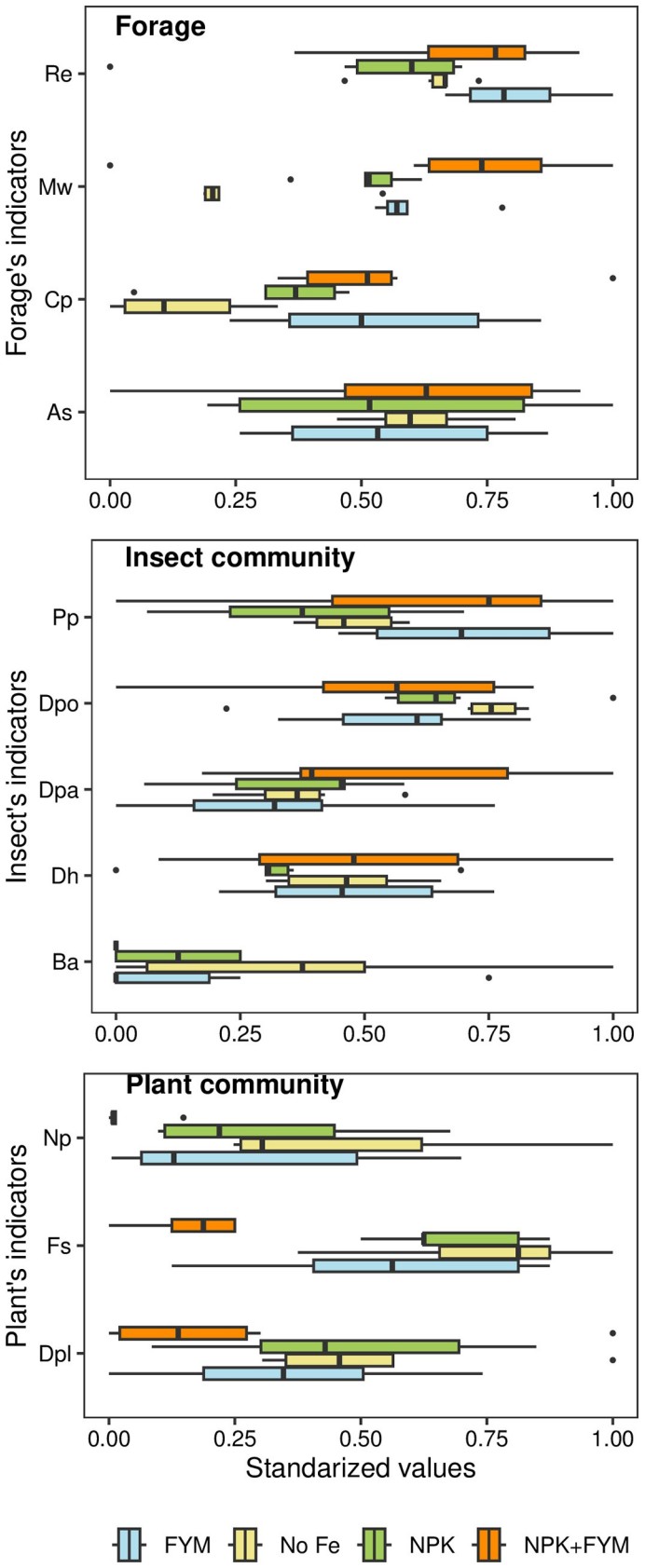

**Fig 2. Standardized values for each of the 12 indicators that describe forage, insect communities and plant communities for each treatment.** Forage: Re: Ruminant metabolizable energy. Mw: Mean dry weight 2011–2014. Cp: Crude protein. As: Ash. Insect community: Pp: Percent of parasitism. Dpo: Shannon-Weiner diversity of pollinators. Dpa: Shannon-Weiner diversity of parasitoids. Dh: Shannon-Weiner diversity of herbivores. Ba Abundance of bumblebees. Plant community: Np: nectar productivity. Fs: Forb species richness. Dpl: Diversity of plants. No Fe: No-fertilizer. NPK: mineral fertilizer (20:10:10), FYM: farm yard manure, NPK+FYM: mineral fertilizer and farm yard manure. N = 24.

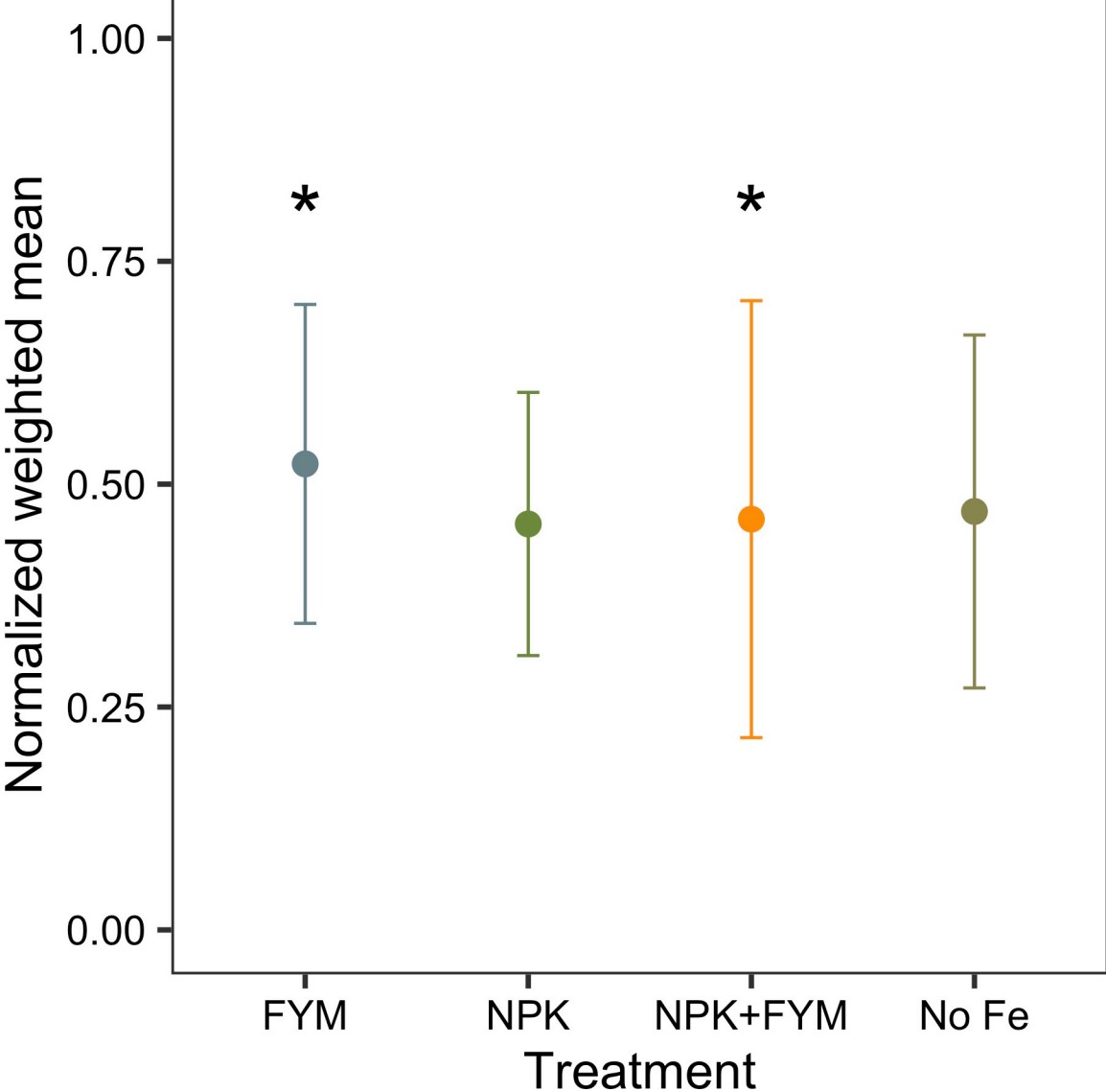

**Fig 3. Optimization of the different indicators under each treatment.** Optimization point is defined as the point where the mean value is maximized with the lowest standard deviation. Under this approach the optimum treatment is the one that minimizes trade-offs between biodiversity and forage production. Among the treatments FYM, showed the higher mean with the lowest standard deviation. Points show standardized weighted mean. Bars: show standardized weighted standard deviation. No Fe: no-fertilizer, NPK: mineral fertilizer (20:10:10), FYM: farm yard manure, NPK+FYM: mineral fertilizer and farm yard manure. * Indicates significant higher values. FYM has a significant higher weighted mean than NPK and No fertilizer, while NPK+FYM has the highest weighted standard deviation across treatments.

the no-fertilizer control. Thus, the application of NPK+FYM shows significantly more variation in the response variables. This variation is due to a high performance in livestock production but low performance in most biodiversity related indicators.

### Estimation of the number of livestock that can be supporting by the different fertiliser treatments

When comparing the stock feeding capacity for each of the treatments, the no-fertilizer control supported the lowest number of cattle and NPK+FYM the highest, with a difference of one animal per hectare of forage between the two treatments (Table 5 in S1 File). However, crude protein content in hay was only adequate for livestock under FYM and FYM+NPK. An important point is that the calculations focus on requirements during the first and second trimester of the cow's pregnancy, and supplements may be needed closer to calving given the modest metabolizable energy contents of the hay.

While fertilizer addition was needed to meet crude protein needed to feed suckler cows during the winter, the hay from the unfertilized control plots appeared to provide adequate nutrition for mixed ewes. Fertilizer treatment influenced the number of sheep that could be potentially fed per hectare of forage, thus the no-fertilizer control vs FYM has a difference of 17 sheep for the control vs 24 sheep for FYM and 26 sheep for NPK+FYM. Finally, NPK-only 23 sheep. However, in all cases, feeding supplements will still be needed to meet increased nutritional demands at key physiological stages (e.g., pregnancy).

## Discussion

In this study, we compared the performance and described the trade-offs among livestock production, plant communities and insect communities, under four different fertilizer treatments in an upland grass forage system managed under agri-environmental schemes. We observed strong trade-offs in all the fertilizer treatments, as all treatments performed well for some indicators and less well for others. Overall, FYM was the fertilizer treatment closest to the optimum point between biodiversity and forage production having the lowest trade-offs between indicators. Finally, we found that after 27 years of no nutrient addition, forage production is predicted to be sufficient to maintain low sheep densities but not sufficient for upland cattle. In what follows, we first discuss the limitations of the work and then discuss the results with further literature.

### Limitations

There are three main limitations to our work. First, our results are conservative as in the experiment the maximum fertilizer application rate was constrained as a condition of receiving support payments from the English & Welsh agri-environmental scheme. Farms not within the scheme can apply fertilizer rates two to four times higher, particularly in permanent grassland [50,53]. However, our results shows that even at low rates of fertilizer addition there are important trade-offs between biodiversity and production, the implication being that trade-offs will worsen at even higher levels of input. Second, we did not calculate livestock production using observations of live animals as our experimental plots are too small to support livestock. Rather we scaled-up the plot-level measurements and linked these to animal production values from the agricultural literature. However, this limitation can also be an advantage as comparisons can readily be made between the different types of livestock used in farming. The third limitation of our work is that the proximity between plots could lead to spill over of species among treatments which might not arise with larger distances between plots. In this sense, our results are conservative as spill over among plots will tend to mask effects.

## Maintaining biodiversity and livestock production under different fertilizer regimes

Our results showed that FYM addition is the regime best able to optimize plant community and insect community diversity while sustaining forage production. This is consistent with the notion that organic fertilizers enhance production but with less impact on biodiversity [54]. Surprisingly, we did not find any significant differences between FYM and the no-fertilizer treatment regarding to some individual indicators related to biodiversity, although forage performance on average was 34% higher in FYM. Interesting, we observed that forage indicators were in average 1.6% lower in FYM than in NPK +FYM. However, the impacts of NPK+FYM on the plant community was 62.5% and 100% higher for bumblebees than FYM despite nitrogen inputs being relatively low compared to intensive grass forage systems. In this sense, producing forage by the addition of FYM seems a more sustainable alternative than the addition of higher fertilizer amounts that have proved to be detrimental to the environment in different experiments [55]. Moreover, incorporating FYM rather than mineral fertilizer constitutes an efficient use of biomass and nutrients produced within the farming system in previous growing seasons and eventually returned as animal waste to be processed below-ground [56].

It is important to highlight that the addition of FYM decreases biodiversity as observed in this study and other studies e.g., [12,57,58], with the negative impacts escalating to higher trophic levels of invertebrates [33]. Moreover, in plots within Colt Park trial where seed addition was carried out to accelerate plant community reassembly and improve nectar productivity, FYM still resulted in reduced plant diversity relative to a complete lack of fertilizer [33,59] indicating that the addition of FYM could slow down the restoration of plant diversity in grasslands [12]. Nevertheless, the fact that percent of parasitism in this treatment is higher than in NPK and NPK+FYM is an important difference. The three fertilizer addition treatments increased abundance of leaf miners and parasitoids [33], but the percentage of leaf miners attacked by parasitoids was higher for FYM suggesting better pest control under this treatment. Overall, while we cannot conclude that FYM is the best practice to maximize biodiversity in our field site, we can conclude that FYM is the optimum treatment because it sustained higher levels of forage productivity alongside simultaneously lower impacts on biodiversity.

Substitution of manure by inorganic mineral fertilizers has contributed to the widespread decrease of plant diversity in grasslands [60]. Nevertheless, some studies consider that low mineral fertilization rates can increase productivity without damaging biodiversity in grasslands [61,62]. We observed that the addition of NPK yields minor improvement in forage production as it does not increase the nutritional content of the hay, but it does have negative impacts on biodiversity as it reduces bumblebee abundance and it may have negative effects on pest control by reducing parasitism. Our results therefore provide no support for introducing low NPK to sustain livestock production in this upland grass forage system [see also 61,63–65]. Indeed, since the critical load for nitrogen for European low and medium altitude meadows is 20–30 kg N ha$^{-1}$ yr$^{-1}$, negative effects on biodiversity are expected even at this low application rate.

Finally, in our field site we did not observe increases in productivity due to higher plant diversity since the unfertilized control had the lowest yield (mean dry weight), which in combination with low crude protein and ruminant metabolize energy were estimated to be insufficient to support beef production. It was suitable for mixed ewes though. One solution to maintain animal production under this type of management is to consider a broader range of livestock, an approach which is been used in some restoration projects where the aim is to increase biodiversity on working farms [66] This type of management requires a shift in perspective; from intensifying the grassland system to meet the production requirements of

livestock with high nutrient demands, toward matching the type of livestock and farming system to the ecosystem to sustain both production and biodiversity. While this approach is unlikely to be appealing to farmers whose goal is to maximize livestock production, it could provide an extra incentive to farmers to join programs aiming to restore and preserve seminatural grasslands [see 67]. These programs, so far, highlight the importance of creating floral resources for pollinators to maintain crop pollination. However, enhancing crop pollination is not a main concern for livestock breeders whose production does not rely directly on pollination. The fact that forage from restored areas is suitable for livestock (albeit only some livestock) after stopping the use of fertilizers could increase farmers' interest in restoring areas within their farm.

Although our study measured responses in relatively small plots, other work has shown that optimization is theoretically possible at landscape scale [27,51]. Yet the strong trade-offs observed in our study suggest that simultaneous maximization of biodiversity and productivity when aiming to restore grassland cannot be achieved within the same management unit. This suggests that delivery will require multi-farm and landscape-scale perspectives [25,51,68]. Further research is needed to test how choosing plant species that are good for insect communities, especially pollinators, together with nutrient rich and productive species would help to maximize livestock productivity and biodiversity in restored grasslands.

## Conclusions

There are strong trade-offs between increasing forage production and conservation of insect biodiversity, these being seen even at the low fertilization levels added under agri-environmental schemes. Maintaining the traditional practice of adding farm yard manure offers the overall best point between the two aims. It has a lower impact on biodiversity than the higher fertilizer treatments, and a higher nutrient content in forage than low mineral fertilization. However, if the aim is to restore plant diversity in grasslands, fertilization should stop being used to keep livestock productivity high. Rather alternative livestock should be considered.

Intensive farming practices around the world have impacted negatively on grassland biodiversity for decades [10]. This has led to a variety of policy responses regulating the impact of agricultural intensification. Despite this though, biodiversity loss continues in agricultural systems. Considering multiple perspectives is essential to implement solutions to environmental problems associated with food production [5]. Moreover, rethinking the typical approach of adapting the vegetation (by fertilizing it) to livestock needs, to one of adapting the livestock type to the existing vegetation, could reduce fertilizer inputs and maintain economically viable production. To move this field of research forward by more than incremental amounts, conservation ecologists and livestock managers need to work collaboratively to devise management approaches that can achieve a meaningful compromise in delivering both insect biodiversity and livestock production. In particular, we need more studies that link biodiversity and farmers' profit margins e.g., [69], as these significantly improve our ability to identify sustainable solutions acceptable to both parties.

## Statement of authorship

EVG conceived the main idea which was developed with JM and SS. SS and EVG conducted the statistical analysis and hay sampling. SS designed the randomization analysis and figure for the randomization analysis. SW did the calculation of hay productivity. MF and EVG performed the livestock production estimation and forage suitability. EVG wrote the manuscript with input from SS, MF and JM. All authors assisted with revisions.

## Supporting information

**S1 Data.**
(XLSX)

**S1 File. Fertilization by manure minimizes the trade-offs between biodiversity and forage production in agri-environment scheme grasslands.**
(PDF)

**S2 File.**
(R)

## Acknowledgments

We would like to thank Natural England for permission to use their site for fieldwork, especially the managers of Ingleborough National Nature Reserve, C. Newlands and A. Hinde. We are grateful to J. Deeming, M. Pavett R. Barnett, E. Clare and C. Godfray for helping in insect identification, E. Grount for field assistance and J. Martinez-Zavaleta for lab assistance. The original manuscript was improved thanks to comments by T. Timberlake and R. Marrs. We are grateful to R. Bardgett for providing information on soil fertility and advice.

## Author Contributions

**Conceptualization:** Edith Villa-Galaviz, Simon M. Smart, Jane Memmott.

**Data curation:** Edith Villa-Galaviz, Susan E. Ward.

**Formal analysis:** Edith Villa-Galaviz, Simon M. Smart.

**Funding acquisition:** Simon M. Smart, Jane Memmott.

**Investigation:** Susan E. Ward.

**Methodology:** Edith Villa-Galaviz, Simon M. Smart, Susan E. Ward, Mariecia D. Fraser.

**Resources:** Susan E. Ward.

**Supervision:** Simon M. Smart, Jane Memmott.

**Validation:** Mariecia D. Fraser.

**Visualization:** Edith Villa-Galaviz, Simon M. Smart.

**Writing – original draft:** Edith Villa-Galaviz.

**Writing – review & editing:** Edith Villa-Galaviz, Simon M. Smart, Susan E. Ward, Mariecia D. Fraser, Jane Memmott.

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
