## [Decision Letter · Decision Letter 0]

31 May 2023

PONE-D-23-09762Fertilization using manure minimizes the trade-offs between biodiversity and forage production in agri-environment scheme grasslandsPLOS ONE

Dear Dr. Villa-Galaviz,

Thank you for submitting your manuscript to PLOS ONE. After careful consideration, we feel that it has merit but does not fully meet PLOS ONE’s publication criteria as it currently stands. Therefore, we invite you to submit a revised version of the manuscript that addresses the points raised during the review process.

We look forward to receiving your revised manuscript.

Kind regards,

Paulo H. Pagliari

Academic Editor

PLOS ONE

Journal Requirements:

   "Species ID was funded by Bristol Centre for Agricultural Innovation (BCAI, project no.42). https://www.bristol.ac.uk/biology/bcai/

EVG was funded by a grant from CONACYT, Mexico for graduate studies (381429). https://conacyt.mx

SMS was supported in part by the UK-SCaPE program delivering National Capability (NE/R016429/1) funded by the Natural Environment Research Council. 

 " ext-link-type="uri" xlink:type="simple">https://www.ukri.org/councils/nerc/" 

   "We would like to thank Natural England for permission to use their site for fieldwork, especially the managers of Ingleborough National Nature Reserve, C. Newlands and A. Hinde. We are grateful to J. Deeming, M. Pavett R. Barnett, E. Clare and C. Godfray for helping in insect identification, E. Grount for field assistance and J. Martinez-Zavaleta for lab assistance. The original manuscript was improved thanks to comments by T. Timberlake and R. Marrs. The project was partly funded by Bristol Centre for Agricultural Innovation (BCAI). EVG was funded by a grant from CONACYT, Mexico for graduate studies (381429). SMS was supported in part by the UK552 SCaPE program delivering National Capability (NE/R016429/1) funded by the Natural Environment Research Council. The authors declare that there is no conflict of interest."

   "Species ID was funded by Bristol Centre for Agricultural Innovation (BCAI, project no.42). https://www.bristol.ac.uk/biology/bcai/

EVG was funded by a grant from CONACYT, Mexico for graduate studies (381429). https://conacyt.mx

SMS was supported in part by the UK-SCaPE program delivering National Capability (NE/R016429/1) funded by the Natural Environment Research Council. 

" ext-link-type="uri" xlink:type="simple">https://www.ukri.org/councils/nerc/"

Additional Editor Comments: 

Please follow the provided comments/suggestions by the reviewers and provide a revised annotated version of the manuscript.

Reviewers' comments:

Reviewer's Responses to Questions

**Comments to the Author**

1. Is the manuscript technically sound, and do the data support the conclusions?

Reviewer #1: Yes

Reviewer #2: Yes

Reviewer #3: Yes

Reviewer #4: Yes

Reviewer #5: Yes

2. Has the statistical analysis been performed appropriately and rigorously? 

Reviewer #1: Yes

Reviewer #2: No

Reviewer #3: Yes

Reviewer #4: Yes

Reviewer #5: Yes

3. Have the authors made all data underlying the findings in their manuscript fully available?

Reviewer #1: Yes

Reviewer #2: Yes

Reviewer #3: Yes

Reviewer #4: Yes

Reviewer #5: Yes

4. Is the manuscript presented in an intelligible fashion and written in standard English?

Reviewer #1: Yes

Reviewer #2: Yes

Reviewer #3: Yes

Reviewer #4: Yes

Reviewer #5: Yes

5. Review Comments to the Author

Reviewer #1: Long-term experiments are a unique source of information and results. They are financially and time consuming. The results are highly desirable from a scientific point of view and provide valuable insights for practice. From this perspective, I appreciate the work of the authors, who have done a wonderful job on an interesting topic. The work is very comprehensive, yet written in a readable and informative manner. The statistical analysis is done very well in R. The answers that came to my mind have been answered in the following text. In fact, I am pleased with the results, as manure is one of the best materials the agricultural (crop) community currently has. Unfortunately (and I mean especially in conventional agriculture) its production is low or on the decline in some countries, leading to a deterioration of soil properties, especially when combined with mineral nitrogen, which is applied at a high rate.

In summary, the article is, in my opinion, very well done (at a high level in fact).

Reviewer #2: The MS was well, my comments were below:

1) The long-term nutrient manipulation experiment was ok, please give the initial soil nutrient before the experiment begin.

2) Please add the organic matter and nutrient contents of farmyard manure.

3) For some figure, please add different letter of a significant difference for all treatments.

4) Please add the conclusion.

Reviewer #3: Dear Authors,

The manuscript is written meticulously and in my view ready for publication with a minor revision. The limitations of the study immediately followed after discussion section. Please see if should be replaced at the end of the article or may be just before conclusion. However, the conclusion section is also missing. Please see if it is required or not.

All the best

Reviewer #4: The work compares indicators of pasture production and biodiversity under various fertilization treatments. To do this, it uses samples that have been carried out in previous years on a long-term trial and incorporates new specific samples. I think the work is interesting and well synthesizes all the information collected during the essay.

General comments:

The abstract could summarize the results or at least mention which indicators were measured.

In the introduction, it would be necessary to present the different fertilizers that will be used. In the last paragraph the objectives are mentioned and it is not clear what the treatments will be.

Many results are presented in the supplementary material. I think that synthetic figures could be elaborated and included in the work. That might avoid having to present the statistical differences in the text. I think the p-values in the text make reading a bit difficult.

In the discussion paragraph that compares indicators between fertilization treatments (452-465) differences between treatments are mentioned. It would be good to indicate the meaning of the differences. For example, instead of saying differences in production were observed between FYM and NPK + FYM, saying NPK + FYM produced slightly (x%) more than FYM.

I think the discussion highlights two main conclusions: 1) that FYM optimizes services and 2) that no fertilization maximizes diversity. The first one has more to do with the objectives (and the methodology) of this work, however, in the end, more emphasis is placed on the second one. I understand that in the end the discussion is on a landscape scale, but it would be good to include a final paragraph with the final conclusions that emerge from this work.

Line 77) replace grazing intensity with reducing grazing intensity

Line 114) compare the performance of four … (without the). Except that tratments are mentioned previously.

Line 137) Check dose of fertilizers. I think there are some problems in the calculations (e.g. N = 12*6 = 72 kg/ha/y)

Line 144) There were four fertilzar treatments. Or no-fertilizad plot were separated?

Line 247) Normalization does not necessarily provides values between 0 and 1.

Reviewer #5: The manuscript discusses a 27-year-long experiment examining the effects of four nutrient inputs on biodiversity and forage production, providing valuable insights into their survival. Since not much is known about how well these treatments work, the current study is especially interesting.

Additional comments:

Add kg to units in abstract.

Statistical analysis: Please elaborate on the specific R package(s) that were utilized.

6. PLOS authors have the option to publish the peer review history of their article (what does this mean?). If published, this will include your full peer review and any attached files.

Reviewer #1: No

Reviewer #2: **Yes: **Kailou LIU

Reviewer #3: No

Reviewer #4: **Yes: **Martin Durante

Reviewer #5: **Yes: **Ali Mokhtassi-Bidgoli

---

## [Author Response · Author response to Decision Letter 0]

10 Jul 2023

Reviewer #1: 

There are no comments to address for Reviewer 1.

Reviewer #2: 

 1) The long-term nutrient manipulation experiment was ok, please give the initial soil nutrient before the experiment begin.

Response: We contacted the researchers in charge of the experiment. Professor Richard Bardgett informed us that tracking such information is hard. However, he also told us that the initial soil nutrient before the experiment begin should be quite similar to the adjacent meadow where the management has remained similar to what it was . We have added such information in the manuscript.

2) Please add the organic matter and nutrient contents of farmyard manure.

R: We added information on dry matter and nutrient contents (N, P and K) in farmyard manure in the methods section. Ln 137-139.

3) For some figure, please add different letter of a significant difference for all treatments.

Response: We have indicated in figures 1 and 3 the significant differences among treatments

4) Please add the conclusion.

Response: we have separated the conclusions from discussion by adding the appropriate subheading. 

Reviewer #3: 

1) The limitations of the study immediately followed after discussion section. Please see if should be replaced at the end of the article or may be just before conclusion. However, the conclusion section is also missing. Please see if it is required or not.

Response: Thank you for the comments and for noticing this missing part. We have separated the conclusions from discussion by adding the appropriate subheading.

Reviewer #4: 

1) The abstract could summarize the results or at least mention which indicators were measured.

Response: we added information of the indicators measured in lines 36-38. We have also summarized the results. 

2) In the introduction, it would be necessary to present the different fertilizers that will be used. In the last paragraph the objectives are mentioned and it is not clear what the treatments will be.

Response: We added the fertilizer treatments in lines 108-109

3) Many results are presented in the supplementary material. I think that synthetic figures could be elaborated and included in the work. That might avoid having to present the statistical differences in the text. I think the p-values in the text make reading a bit difficult.

Response: We added a new table with the p values of the randomizations for the indicators when compared individually into the main text. This values were previously reported in the supp. Info. We deleted the p values from the paragraph and refer reader to the table. We also removed those results from the supplementary information. Likewise, we also make it clearer the reading of the results in different sections, to reduce p vales being distracting at reading. In the discussion paragraph that compares indicators between fertilization treatments (452-465) differences between treatments are mentioned. It would be good to indicate the meaning of the differences. For example, instead of saying differences in production were observed between FYM and NPK + FYM, saying NPK + FYM produced slightly (x%) more than FYM.

Response: We have added the percentages in the paragraph. Ln 448-450

4) I think the discussion highlights two main conclusions: 1) that FYM optimizes services and 2) that no fertilization maximizes diversity. The first one has more to do with the objectives (and the methodology) of this work, however, in the end, more emphasis is placed on the second one. I understand that in the end the discussion is on a landscape scale, but it would be good to include a final paragraph with the final conclusions that emerge from this work.

Response: Thank you for the comment we have added a paragraph with the conclusions starting in line 536.

5) Line 77) replace grazing intensity with reducing grazing intensity

Response: Thank you for the observation. We changed the wording, see ln 75.

6) Line 114) compare the performance of four … (without the). Except that tratments are mentioned previously.

Response: Thank you, following previous recommendation we added the fertilizer treatments earlier in the paragraph.

7) Line 137) Check dose of fertilizers. I think there are some problems in the calculations (e.g. N = 12*6 = 72 kg/ha/y)

Response: We have corrected the calculations in the method sections and throughout the manuscript. 

Line 144) There were four fertilzar treatments. Or no-fertilizad plot were separated?

Response: Thank you, we have edited the manuscript, so it is clear that there were four fertilizer treatments. Ln 136.

Line 247) Normalization does not necessarily provides values between 0 and 1.

Response: Thank you. We have replace this sentence as follows: “this providing values in the same unit for all indicators while retaining the distributional features of each indicator measured”.

Reviewer #5: 

1) Add kg to units in abstract.

Response: Thank you. We have added the missing information.

2) Statistical analysis: Please elaborate on the specific R package(s) that were utilized.

Response: Thank you for noting it. We added the packages and now the text reads as follows:

“Analyses used the vegan (40), dtplyr (41), plyr (42), and ggplot2 (43) packages R version 4.1.3 (44)”

---

## [Decision Letter · Decision Letter 1]

17 Aug 2023

Fertilization using manure minimizes the trade-offs between biodiversity and forage production in agri-environment scheme grasslands

PONE-D-23-09762R1

Dear Dr. Villa-Galaviz,

We’re pleased to inform you that your manuscript has been judged scientifically suitable for publication and will be formally accepted for publication once it meets all outstanding technical requirements.

Kind regards,

Paulo H. Pagliari

Academic Editor

PLOS ONE

Additional Editor Comments (optional):

Reviewers' comments:

Reviewer's Responses to Questions

**Comments to the Author**

1. If the authors have adequately addressed your comments raised in a previous round of review and you feel that this manuscript is now acceptable for publication, you may indicate that here to bypass the “Comments to the Author” section, enter your conflict of interest statement in the “Confidential to Editor” section, and submit your "Accept" recommendation.

Reviewer #3: (No Response)

Reviewer #5: All comments have been addressed

2. Is the manuscript technically sound, and do the data support the conclusions?

Reviewer #3: Yes

Reviewer #5: Yes

3. Has the statistical analysis been performed appropriately and rigorously? 

Reviewer #3: Yes

Reviewer #5: Yes

4. Have the authors made all data underlying the findings in their manuscript fully available?

Reviewer #3: No

Reviewer #5: Yes

5. Is the manuscript presented in an intelligible fashion and written in standard English?

Reviewer #3: Yes

Reviewer #5: Yes

6. Review Comments to the Author

Reviewer #3: (No Response)

Reviewer #5: (No Response)

7. PLOS authors have the option to publish the peer review history of their article (what does this mean?). If published, this will include your full peer review and any attached files.

Reviewer #3: No

Reviewer #5: **Yes: **Ali Mokhtassi-Bidgoli

---

## [Editor Report · Acceptance letter]

25 Sep 2023

PONE-D-23-09762R1 

Fertilization using manure minimizes the trade-offs between biodiversity and forage production in agri-environment scheme grasslands 

Dear Dr. Villa-Galaviz:

I'm pleased to inform you that your manuscript has been deemed suitable for publication in PLOS ONE. Congratulations! Your manuscript is now with our production department. 

Kind regards, 

on behalf of

Dr. Paulo H. Pagliari 

Academic Editor

PLOS ONE